# Investigation of Salmonella Phage–Bacteria Infection Profiles: Network Structure Reveals a Gradient of Target-Range from Generalist to Specialist Phage Clones in Nested Subsets

**DOI:** 10.3390/v13071261

**Published:** 2021-06-28

**Authors:** Khatuna Makalatia, Elene Kakabadze, Nata Bakuradze, Nino Grdzelishvili, Ben Stamp, Ezra Herman, Avraam Tapinos, Aidan Coffey, David Lee, Nikolaos G. Papadopoulos, David L. Robertson, Nina Chanishvili, Spyridon Megremis

**Affiliations:** 1Eliava Institute of Bacteriophage, Microbiology and Virology, Tbilisi 0162, Georgia; k.makalatia@gmail.com (K.M.); elene.kakabadze@pha.ge (E.K.); nata.bakuradze@pha.ge (N.B.); n.grdzelishvili@pha.ge (N.G.); 2Faculty of Medicine, Teaching University Geomedi, Tbilisi 0114, Georgia; 3MRC-University of Glasgow Centre for Virus Research, University of Glasgow, Glasgow G61 1QH, UK; b.stamp.1@research.gla.ac.uk (B.S.); David.L.Robertson@glasgow.ac.uk (D.L.R.); 4Department of Biology, University of York, Wentworth Way, York YO10 5DD, UK; elh605@york.ac.uk; 5Division of Evolution and Genomic Sciences, The University of Manchester, Manchester M13 9GB, UK; avraam.tapinos@manchester.ac.uk; 6Department of Biological Sciences, Munster Technological University, T12 P928 Cork, Ireland; aidan.coffey@cit.ie (A.C.); david.lee@mycit.ie (D.L.); 7Division of Infection, Immunity and Respiratory Medicine, The University of Manchester, Manchester M13 9PL, UK; nikolaos.papadopoulos@manchester.ac.uk; 8Allergy Department, 2nd Paediatric Clinic, National and Kapodistrian University of Athens, 115 27 Athens, Greece

**Keywords:** salmonella, bacteriophages, virus, bacteria, nestedness, modularity, infection, network, evolution, speciation

## Abstract

Bacteriophages that lyse Salmonella enterica are potential tools to target and control Salmonella infections. Investigating the host range of Salmonella phages is a key to understand their impact on bacterial ecology, coevolution and inform their use in intervention strategies. Virus–host infection networks have been used to characterize the “predator–prey” interactions between phages and bacteria and provide insights into host range and specificity. Here, we characterize the target-range and infection profiles of 13 Salmonella phage clones against a diverse set of 141 Salmonella strains. The environmental source and taxonomy contributed to the observed infection profiles, and genetically proximal phages shared similar infection profiles. Using in vitro infection data, we analyzed the structure of the Salmonella phage–bacteria infection network. The network has a non-random nested organization and weak modularity suggesting a gradient of target-range from generalist to specialist species with nested subsets, which are also observed within and across the different phage infection profile groups. Our results have implications for our understanding of the coevolutionary mechanisms shaping the ecological interactions between Salmonella phages and their bacterial hosts and can inform strategies for targeting Salmonella enterica with specific phage preparations.

## 1. Introduction

Salmonella enterica serovars Enteritidis and Typhimurium are major foodborne pathogens of worldwide concern, which often cause severe diarrheal diseases sometimes with fatal outcomes. According to the WHO report, 550 million people are infected annually, including 220 million children under the age of five [1]. The majority of these cases of domestically acquired salmonellosis are caused by various Salmonella serovars transmitted through the food chain (The US Centers for Disease Control and Prevention) [2]. Increasing antibiotic-resistance of this bacterium is aggravating the epidemic situation [3]. Due to the increasing problem of antibiotic resistance, the development of new strategies to sustainably control food-borne pathogens is urgently needed. Bacteriophages (phages) are potential alternative tools to target Salmonella infection. In post-Soviet Union countries, phage preparations have a long history of application for treatment and prophylaxis against dysenterial diseases, such as shigellosis, escherichiosis, and salmonellosis. These preparations have been successfully used for prophylaxis of salmonellosis among civilian population as well as in Red Army units [4,5,6]. In Western countries phage applications are used for biocontrol in foods [3]. Better understanding of phage–bacteria interactions will facilitate the development of a more rational approach to select appropriate phages for therapy, prophylaxis or biocontrol of Salmonella infections. Therefore, investigating the host range of Salmonella phages is a key to understand their impact on bacterial ecology, coevolution and inform intervention strategies.

One way to study a phage’s host ranges and ecological interactions is through the investigation of phage–bacteria infection networks [7,8,9] (PBINs). Essentially, these networks (or graphs) represent predator–prey interactions between phages and bacteria, respectively. These ecological relationships are graphed as a bipartite network where edges connect nodes from two different subsets, i.e., phages and bacteria. In such networks, an interaction between a single bacterium and a phage could have an impact on other members of the network too. Studying the structure of PBINs essentially addresses the question of “who-kills-whom” and provides important information regarding the ecological, biological and coevolutionary mechanisms underlying the network’s structure [10,11,12,13,14]. For example, large variation in host range species suggests that some phages have evolved to infect many hosts (generalists with wide host ranges), while others have evolved to infect a few select hosts (specialists with narrow host ranges). Equally, large variation in susceptibility host species suggests that the network consists of hosts with generalist and specialist resistance repertoires. At an evolutionary scale, large variation in host and susceptibility range sizes could indicate that particular modes of coevolution are occurring [10,15].

In this study, we investigate the in vitro infection profiles of different Salmonella phages against a wide range of Salmonella bacterium strains. We further use these data to construct a Salmonella phage–bacteria infection network (PBIN) and study its structure. Based on the above, we define a set of different Salmonella phage infection groups (or clusters) and explore their association to known Salmonella phage characteristics. Our aim is to better understand the underlying ecological and evolutionary dynamics that shape the Salmonella phages’ host-range.

## 2. Materials and Methods

### 2.1. Phages and Bacterial Strains

Thirteen Salmonella-specific individual phage clones previously isolated from environmental sources (sewage, the river and sea water samples, milk) [16,17] were used for testing the Salmonella strains for susceptibility, these are GEC_vB_B1, GEC_vB_B3, GEC_vB_NS7, GEC_vB_BS, GEC_vB_MG, GEC_vB_7A, GEC_vB_N5, GEC_vB_N8, GEC_vB_N3, GEC_vB_M4, GEC_vB_M5, GEC_vB_HIL, and GEC_vB_TR. The genomes of eight out of the above phages were previously sequenced and annotated by short-read high throughput sequencing [16] (MiniSeq Illumina NGS platform, Illumina, San Diego, CA, USA) (Appendix A). Six genomes have been deposited in NCBI and are available via GenBank accession numbers in Appendix A. Five of these phages belong to the Myoviridae family with two different genera: Felixounavirus (GEC_vB_B1, GEC_vB_B3, GEC_vB_NS7, and GEC_vB_BS), and Seunavirus (GEC_vB_MG). Three phages belong to the Demerecviridae family of the genus Tequintavirus (GEC_vB_N3, GEC_vB_N5, GEC_vB_N8). The second group of phages, sequenced using long-read sequencing [18] (MinION, Oxford Nanopore Technology, ONT, Oxford, UK), is composed of five members belonging to Siphoviridae family, genus Jerseyvirus (GEC_vB_M4, GEC_vB_M5, GE_vB_HIL), Myoviridae family genus Felixounavirus (GEC_vB_7A), and Podoviridae family genus Lederbergvirus (GEC_vB_TR) (Appendix A).

Altogether, 141 Salmonella strains originating from Ireland [19] were included in this study. These strains are related to various serovars: S. Typhimurium (*n* = 40), S. Dublin (*n* = 22), S. Enteritidis (*n* = 21), S. Anatum (*n* = 10), S. Infantis (*n* = 8), S. Newport (*n* = 7), S. Bredeney (*n* = 5), S. Derbey (*n* = 4), S. Braenderup (*n* = 2), S. Germinara (*n* = 2), S. Uganda (*n* = 2), S. Senftenberg (2), S. Kentucky (*n* = 1), S. Java (1), S. Branderburg (*n* = 1), S. Bareilly (*n* = 1), S. Virchow (*n* = 1), S. Goldcost (*n* = 1), S. Poona (*n* = 1), and unknown serotypes (*n* = 9). Overall, 19 serotypes of Salmonella were included into the work. The majority of these strains (*n* = 88) were of veterinary origin (bovine *n* = 36, porcine *n* = 34, poultry *n* = 17, duck *n* = 1), 27 were isolated from humans, 6 strains were obtained from food products (cheese, vegetables, fish) and 1 from terpene.

### 2.2. Antibiotic Susceptibility Tests

The antibiotic resistance profile of all isolates was determined using Kirby–Bauer method applied to 16 antibiotics: Ampicillin (A, 10 µg/disk), Amikacin (Ak, 30 µg/disk), Cefotaxime (Ctx 30 µg/disk), Chloramphenicol (C, 30 µg/disk), Ciprofloxacin (Cip, 5 µg/disk), Enrofloxacin (Enr, 5 µg/disk), Gentamycin (Cn, 30 µg/disk), Lomefloxacin (Lom, 10 µg/disk), Nalidixic acid (Nal, 30 µg/disk), Norfloxacin (Nor, 2 µg/disk), Spectinomycin (Sp, 10 µg/disk), Streptomycin (S, 10 µg/disk), Sulfonamides (S3, 300 µg/disk), Teicoplanin (Tec, 30 µg/disk), Tetracycline (Te, 30 µg/disk) and Trimethoprim (W, 5 µg/disk). Antibiotic susceptibility, intermediate resistance or resistance was determined according to NCCLS standards [20].

Phage isolation and preparation of high titter phage stocks were performed as described in Makalatia et al., 2020 [16].

### 2.3. Bacteriophage Susceptibility Test

Assessment of phage activity against different bacterial strains was performed using the double layer agar method [16]. Briefly, overnight bacterial cultures and phage stocks were diluted in the sterile lysogeny broth (LB) (Oxoid Limited, Basingstoke, UK) to final concentrations of 10^8^, 10^7^, and 10^6^ pfu/mL, respectively. Bacterial lawns were made on pre-prepared LB agar (2%) plates using a 300 µL of each bacterial test cultures (titer 10^8^ cfu/mL) mixed with 5 mL of the soft LB agar (0.7%) and air-dried for 10–15 min; 5 µL of each phage clone and cocktail was applied on each plate. Thus, the ratio between the infecting phage particles and bacterial cells, i.e., the multiplicity of infection (MOI) were equal to 1.0, 0.1, and 0.01. The plates were incubated at 37 °C for 18 h and results were recorded. Phage activity was assessed based on visualization. All types of lyses (confluent lysis (CL), semi-confluent lysis (SCL), opaque lysis (OL), countable number of phage plaques on the phage application spots (“taches vierges”, TV) were considered as positive results designated as “S” (sensitive). Uninterrupted bacterial growth on the spot was recorded as “R” (resistant).

### 2.4. Representation of In Vitro Infection Data in a Phage–Bacterium Infection Matrix

Successful infection experiments were translated to a binary dataset, with any evidence of lysis becoming ‘True’, represented as 1, and phage–host pairs annotated as resistant (score 0) remaining as 0 to represent ‘False’. The Salmonella dataset has some incomplete rows (hosts that have not been tested against the complete set of phages), which were discarded. Any hosts showing no evidence of lysis by any of the phage (rows in the final matrix with only 0 s) were discarded. For the Salmonella dataset, 7 out of 148 hosts (Armenian origin) were discarded due to missing data and a further 8 hosts were discarded due to showing no lysis by any of the phage isolates. The Salmonella phage–bacteria infection matrix with accompanying metadata can be found in the Appendix A.

### 2.5. Assignment of Infection Profiles and Clustering

The Jaccard index was used to describe similarity between phages based on the bacterial strains they could infect, i.e., based on their infection profiles, and vice versa. The Jaccard distances (dissimilarity) between phages and bacteria were annotated on principal components. To identify groups of phages and bacteria based on similar infection profiles, unsupervised agglomerative hierarchical clustering (complete linkage) of the Jaccard score was performed. The cluster membership of the Salmonella phages and bacteria were further annotated on the principal components. Principal component decomposition was performed to monitor the percentage of variance explained in different components (dimensions). The Jaccard similarity, Jaccard distances, agglomerative hierarchical clustering, and principal component analysis were performed in MATLAB R2018a. The Salmonella phage taxonomy tree was produced using PhyloT v2 and iTOL v6 (a phylogenetic tree generator, based on NCBI or GTD taxonomy) [21,22].

### 2.6. Construction of Salmonella Phage–Bacterium Infection Network

The Salmonella phage–bacterium infection matrix (PBIM) was transformed into a bipartite network (PBIN) [10]. Nodes represented Salmonella phages and bacteria connected through edges. A basic description of the network is presented in Appendix A. PBIN visualization and annotation was performed using NAViGaTOR [23]. Phage nodes are presented as polygons and bacterial nodes as spheres. Node and edge colors represent phage infection profile groupings as described above. The network was arranged using the GRIP layout (Graph Drawing with Intelligent Placement) [23]. Clique assignment was performed using NAViGaTOR [23]; a clique is the strictest possible definition of a graph cluster. In a k-clique, k vertices are fully connected i.e., there is an edge between every pair of distinct vertices. Cliques have an edge density of 1.

### 2.7. Calculating Nestedness and Modularity

Nestedness was quantified by the NODF metric, calculated using the R package vegan [24,25]. Modularity was calculated using Barber’s modularity, implemented and optimized by the R package bipartite [26,27,28]. Nestedness and modularity scores were compared to null models Sim1, Sim8 and Curveball. Sim1 and Sim8 were implemented using the R package EcoSimR [29], while Curveball was implemented using the R function provided by Strona et al. [30]. For each dataset, 1200 matrices were simulated for each null model. Nestedness and modularity scores were computed for each of these null matrices, producing a distribution of scores. These distributions can be compared to the scores for the observed Salmonella PBIN. A threshold for nestedness and modularity scores can then be defined, above which less than 5% of null matrix scores fall. Scores for observed data beyond this threshold can be annotated as significant (*p* < 0.05, one-tailed). Briefly, Sim1 assigns equal probabilities of infection to each cell (phage–host pair); Sim8 assigns probabilities proportional to row and column totals; Curveball performs sequential swapping of cell values, preserving row and column totals. These null models can be thought of as a range of stringency: Sim1 being least stringent and making the fewest assumptions of the data, and Curveball being most stringent. Simulation of null matrices and computation of nestedness and modularity scores was performed in parallel using the R packages doParallel [31]. The R code for the nestedness and modularity analyses can be found in the “SalmonellaPBIN.Rmd” file in the Appendix A. The input data can be found in the “salmonellaPBIN.csv” file in the Appendix A.

### 2.8. Statistical Analysis

The distribution of variables was tested using the Shapiro–Wilk test. The non-parametric Friedman test was used to compare repeated measures between phage groups or phages. Pairwise comparisons of multiple variables were corrected using the Dunn’s multiple comparison test. Statistical significance was assigned below 0.05. The pairwise comparisons amongst the 13 Salmonella phages are described in the Appendix A. Statistical tests and graphical visualizations were produced in GraphPad Prism 9. The correlation of distance matrices was tested using the Mantel test (9999 permutations). The code is available in R (https://www.rdocumentation.org/packages/ade4/versions/1.7-16/topics/mantel.rtest) (accessed 10 April 2021).

## 3. Results

We analyzed the infection profiles of 13 Salmonella phage species against a set of 141 bacterial strains related to various Salmonella serovars (Appendix A). The Salmonella bacteria strains demonstrated different resistance patterns. The vast majority of isolates had resistance to at least one antibiotic. Multiresistance (resistance to two or more families of antibiotics) was also common. A great number of isolates had resistance to at least three antibiotics with the majority showing a penta-resistance profile. The most common resistance profile was ACSSuTSh, which was mainly associated with Salmonella Typhimurium, many of which have additional resistance to Trimethoprim (ACSSuTTmSh). Resistance to the quinolones was relatively low. Nine had resistance to Nalidixic acid. Only three isolates expressed resistance to the third-generation cephalosporin Cefotaxime. These three isolates were Salmonella Newport [19].

### 3.1. The Salmonella Phage Target-Range Varies between Clones of Different Taxonomic Lineages and Isolated from Different Environmental Sites

The Salmonella phage clones were isolated from four different environmental sources: water samples from the river Mtkari (*n* = 8), the Black Sea (*n* = 3), the artificial Tbilisi Sea (*n* = 1), and a raw cow milk sample (*n* = 1) (Appendix A) (Figure 1a). Based on the isolate source we observed a broad range of phage targets (Figure 1a). The broad target range of multiple phages isolated from the same environmental source suggested that there is a co-occurrence of generalist and specialist species in the same environmental site (Table 1). For example, phage clones from the Mtkvari river lysed 52 (GE_vB_7A) to 111 (GE_vB_B1) bacterial strains, and clones from the Black Sea lysed 33 (GE_vB_M4) to 107 (GE_vB_BS) Salmonella strains (Table 1). The Salmonella phages were taxonomically clustered in four families and six genera (Table 1). Phage clones of the Myoviridae family (*n* = 6) and a subset of them, classified as subfamily Ounavirinae (*n* = 4), infected on average the highest number of different Salmonella bacterial strains and serotypes (mean: 88, 95%CI: 52–111, and mean: 105, 95%CI: 97–111 strains, respectively) (Figure 1b,c). Based on the phage taxonomic cladogram, phage species of clade 1 infected the highest number of bacteria (mean: 105, 95%CI: 96–115) and clade 2 species the lowest (mean: 54, 95%CI: 29–79) (Figure 1d,e). Notably, species of both clades 1 and 2, belonged to the Myoviridae family, however, clade 1 species belonged to the rank of subfamily Ounavirinae; genus Felixounavirus, whereas clade 2 species belonged to the family Myoviridae; genera Felixounavirus and Seunavirus, suggesting a diversification of Salmonella phage target-range within the Myoviridae family. Similar variability was observed within the family of Siphoviridae with species of the Jerseyvirus genus targeting a broad range of Salmonella bacterial strains (*n* = 3, range: 33–82). The target range for phages in the family Demerecviridae of the Tequintavirus genus (clade 3) was lower (*n* = 3, range: 76–96) potentially being influenced by the isolate source which was the same for all clade 3 species (the samples from the river Mtkari).

### 3.2. The Salmonella Phages Are Grouped into Clusters with Different Infection Profiles

The phages with the broadest range of Salmonella targets were GE_vB_B1 which infected 111 Salmonella strains (79%), GE_vB_BS infected 107 strains (76%), and GE_vB_B3 that infected 106 strains (75%). The phages with the narrowest range of bacterial targets were GE_vB_M4 that infected 33 strains (23%) and GE_vB_M5 and GE_vB_7A that both infected 52 Salmonella strains (37%) (Table 1). To gain a better understanding of target-range and specificity, the Jaccard distance was used to measure the pairwise similarity amongst phages based on the bacteria that were able to infect, and vice versa (Figure 2a). Based on these distances, phages and bacteria were annotated in principal components (Appendix A). We observed a divergence of the infection profiles for phages GE_vB_MG, GE_vB_M4, and GE_vB_M5 (Appendix A), suggesting differences in their infection profiles compared to the rest of the phages. Unsupervised Agglomerative hierarchical clustering (complete linkage) was used to identify groups of phages, and bacteria, with similar infection profiles (Figure 2b,c); phages were grouped into four infection profile clusters (Table 1) (Figure 3a–d). The Salmonella bacterial strains were grouped into 21 clusters (Appendix A). Phage cluster 1 contained generalist species that infected on average 100 Salmonella bacterial strains (95%CI: 82–111), equal to about 71% of total Salmonella strains tested, followed by phages of cluster 3 (58%, mean: 83, 95%CI: 75–96), cluster 2 (39%, mean: 55, 95%CI: 52–57), and cluster 4 (30%, mean 43, 95%CI: 33–52) (Figure 3e). About 36% (*n* = 48) of Salmonella bacterial strains were infected by at least one phage from each phage infection cluster (Figure 3f). In phage clusters 1 and 2 we observed the co-clustering of phage clones from different environmental sources, taxonomic families and genera (Table 1), whereas clusters 3 (river Mtkari; Demerecviridae; Tequintavirus) and 4 (Black Sea; Siphoviridae; Jerseyvirus) were more homogenous (Table 1). Therefore, no clear association between phage target-range, environmental source, taxonomy and infection profiles alone was observed.

To further investigate phage speciation, we analyzed the phage infection profiles considering their targets’ taxonomic group at species level, environmental source (isolate type), and infection clusters (bacterial groupings based on hierarchical clustering). Comparison of the absolute count of infected bacterial strains belonging to different bacterial species (serotypes) we observed significant differences (ANOVA Friedman statistic = 26.71, *p* < 0.0001) between phage clusters 1 and 4 (Dunn’s multiple comparison test adjusted *p* = 0.0004, Rank sum diff. 33.5) and clusters 3 and 4 (Dunn’s multiple comparison test adjusted *p* = 0.002, Rank sum diff. 30.0) (Figure 4a). The number of infected bacterial strains with unknown serotype was highest in clusters 1 and 2 (*n* = 8), followed by cluster 3 (*n* = 5), and cluster 4 (*n* = 4). Phages grouped in infection clusters 1 and 3 demonstrated the widest diversity of targets, infecting 19 out of 20 different known bacterial species followed by cluster 2 (14 out of 20) and cluster 4 (9 out of 20) (Figure 4a). These findings suggest that phage clones from clusters 1 and 3 infect a significantly higher number of bacterial strains and from a more diverse pool of bacterial serotypes compared to cluster 4. As a result, the compositional infection profiles of phages in cluster 4 seem to diverge from the other phages (Figure 4b).

Similar differences (ANOVA Friedman statistic = 19.34, *p* = 0.0002) were observed between phage clusters 1 and 4 (Dunn’s multiple comparison test adjusted *p* = 0.0012, Rank sum diff. 22.5), and clusters 3 and 4 (Dunn’s multiple comparison test adjusted *p* = 0.023, Rank sum diff. 17.5) based on the bacterial isolate source (Figure 4c); phages grouped in clusters 1 and 3 successfully infected 9 out of 10 known Salmonella isolate types, followed by cluster 2 phages (8 out of 10) and cluster 4 phages (5 out of 10) (Figure 4d). With respect to bacterial groups based on their infection profiles, significant differences (ANOVA Friedman statistic = 18.30, *p* = 0.0004) were observed between phage clusters 1 and 4 (Dunn’s multiple comparison test adjusted *p* = 0.0006, Rank sum diff. 25.5) (Figure 4e); cluster 1 and 2 phages infected Salmonella strains from 12 out of 21 bacterial infection clusters, cluster 3 phages infected 11 out of 21, and cluster 4 phages infected 4 out of 21 (Figure 4f). No significant correlation was observed between the number of phages in each phage cluster and the number of different bacterial species, bacterial isolates or bacterial clusters (Spearman correlation *p* < 0.05), suggesting that our observations are not influenced by the number of phages within each phage infection cluster. Comparisons of the infection profiles amongst each phage species are described in Appendix A and statistical testing in Appendix A. Collectively, our analysis suggests that these Salmonella phages exhibit a hierarchical distribution based on their target-range (generalist to specialist). This distribution is also evident in the infection profile clusters. The phages grouped in cluster 1 are able to infect a high number of different Salmonella strains including a diverse set of bacterial members from different Salmonella serotypes, isolate type and infection profiles. In contrast, phages grouped in the phage infection cluster 4 are much more specific targeting a certain subset of Salmonella strains.

### 3.3. The Genetic Distances of the Salmonella Phages Are Positively Associated to Their Infection Profiles

To further investigate the hierarchical target range of the phages, we hypothesized that there is a possible association between the heterogeneity of the phage infection profiles and the genetic distances between the phages. To address this, we performed phage whole genome sequence alignments of the available phage complete genome references in NCBI (12 out of the 13 phages) and calculated the genetic distances (Jukes–Cantor and *p*-distance) between all possible phage pairs. A hierarchical clustering and principal component annotation of the phage genetic distances are described in Appendix A. We then tested the association between the genetic and the infection-profile distances described above. We observed a positive association (Jukes–Cantor r 0.14, P distance r 0.17) between the matrix entries, suggesting that smaller differences in infection profiles are generally seen among pairs of phages that are genetically close to each other than far from each other.

### 3.4. The Salmonella Phage–Bacteria Infection Network Is Nested

We represented the Salmonella phage–bacterium infection matrix as a bipartite phage–bacterium infection network (PBIN) (Appendix A), where edges can connect only nodes from two different subsets, i.e., phages and bacteria, and analyzed the structure of the network (see Section 2). Interestingly, the topological organization of the Salmonella network reflects the divergence of specific phage clones based on their infection profiles, in line with the topological organization of the Jaccard distances in the coordinate system, suggesting that PBINs can capture differences in the phage infection profiles. PBINs can be characterized by four different structures (or patterns) (example in Figure 5a) [10]; PBINs can share features of modularity and/or nestedness, i.e., interactions happening between certain sets of bacteria and phages with no interactions across different sets (modular) and/or a range of specialist to generalist bacteria and phages, with interactions forming nested subsets (nested) [8,11]. Visual inspection of the rearranged Salmonella PBIM and PBIN suggested that the networks organization was not random but possibly nested (Figure 5b,c). Nestedness was calculated using NODF (see Section 2). Modularity was calculated using Barber’s modularity, with optimization by the method of Beckett [28] (see Section 2). Nestedness and modularity scores were compared to null models Sim1, Sim8 and Curveball. We find that the Salmonella PBIN is significantly nested when compared to both Sim1 and Sim8 (NODF = 73.76, *p* < 0.001), but not when compared to the Curveball null matrices (Figure 5d). We also observed that the Salmonella PBIN is significantly modular compared to all three null models (Q = 0.1514, *p* < 0.001). However, this modularity score may not be practically convincing, as many interactions are seen outside of the computed modules (Figure 5e). To further investigate this, we followed a network-based approach, and asked if we could observe cliques within the network, i.e., an n-clique of an undirected graph is a maximal subgraph in which every pair of vertices is connected by a path of length n or less. We identified four cliques of phage and bacteria species suggesting that some level of modularity was present (Appendix A).

Overall, we show that the Salmonella PBIN has a non-random structure. The infectivity patterns suggest a gradient of target-range from generalist to specialist phage species with nested subsets and weak modularity, both of which are also evident in the phage infection profiles.

## 4. Discussion

In this study we characterized the target-range of a taxonomically diverse set of Salmonella phages isolated from different environmental sources. Based on the bacteria the phages could infect in vitro, they were robustly assigned to four groups. No clear association between the environmental source, taxonomy and infection profiles was observed, suggesting that none of these factors alone can explain the heterogeneity in infection preferences. This is in line with the recently reported data by Gencay et al. on Salmonella phages isolated from animal, environmental and wastewater samples where phage genus or receptor alone explained less than 50% of variance [32]. Interestingly, we observed a positive association between the genetic distances of the phages and their infection profiles. Genetically related phages often shared more similar infection profiles compared to the distantly related ones, suggesting that the phage genomic composition is linked to target preferences.

Phages assigned to different clusters demonstrated differential infection profiles. Specifically, phages of clusters 1, 3 and 4 infected a variable amount of Salmonella strains from different serotypes, environmental isolates, and bacterial groups. Phage clusters 1 and 2 were quite similar based on the bacterial composition of their infection profiles, whereas phage cluster 4 was the most dissimilar compared to the other three clusters. Notably, the highest variability amongst phage clones was observed when considering the serotype composition of their infection profiles, rather than the environmental source. Indeed, the phages originated from the Georgian environmental samples lysed Salmonella strains isolated from diverse sources including human, veterinary, and food in Ireland. Moreover, in previous studies we have demonstrated that these phages can successfully lyse genetically diverse MDR strains of the S. enterica ssp. enterica isolated from patients in Armenia and Georgia [33,34], as well as Spectrum β-Lactamase (CTX) producing S. enterica Serovar Typhi strains from the Democratic Republic of the Congo [18].

Notably, we observed phage stratification based on host-range. This was evident within most phage clusters (generalist to specialist clones) and amongst the clusters themselves. For example, phage clones in cluster 1 infected on average the highest number of bacterial strains followed by clusters 3, 2 and 4. This suggests that even though there are different infection profiles shared to some extent amongst the different phage clones, the capacity to infect a high (generalist) or low (specialist) number of bacterial strains may be a specific property of a phage (see below). The high effectiveness of cluster 1 phages may be explained by the fact that three out of its five members belong to Myoviridae family genus Felixounavirus, in particular the phages GEC_vB_B1, GEC_vB_B3 and GEC_vB_NS7. Myoviridae phages correspond to phages with contractile tails that have been reported as strictly lytic [35], and Felixounaviruses are known to be virulent representatives of Salmonella phages [36,37], possibly due to their receptors located on the O-antigen and core polysaccharide portions of the LPS [38,39,40]. Therefore, the attachment of bacteriophages to specific receptors of the host bacteria may be a deterministic factor for the extent of the observed target-range. Phage cluster 3 was mainly composed from representatives of the family Demerecviridae genus Tequintavirus (GEC_vB_N3, GEC_vB_N5 and GEC_vB_N8). Genomic analysis revealed that the phages GEC_vB_N5 and GEC_vB_N8 are closely related. Phage GEC_vB_N5 has 94.69% sequence similarity to phage T5 [16,41,42,43], which has both a receptor-binding tail protein Pb5 and a L-shaped tail fiber protein on the phage tail that targets the host outer membrane protein and LPS, respectively [43,44]. Cluster 2 phages are related to the Myoviridae genera Seunavirus and Felixounavirus (GEC_vB_MG and GEC_vB_7A), with the exemption of phage GEC_vB_TR of the Podoviridae family, genus Lederbergvirus. This is the only temperate phage amongst the 13 used in this project. In general, the representatives of Podoviridae are known to be temperate having narrow host ranges. Finally, cluster 4 included two phages of the Siphoviridae family, genus Jerseyvirus (GEC_vB_M4, GEC_vB_M5), which appeared to be closely related [16]. These viruses were part of the same clique in the Salmonella PBIN suggesting possible coevolution leading to increased speciation and narrow target-range (see below). Interestingly, the phage from phage cluster 1 with the narrowest target range is also a Jerseyvirus of the Siphoviridae family (GEC_vB_HIL).

Very little is known about the ecological interactions between Salmonella bacteriophages and bacteria, such as the general structure of infection and resistance patterns between them. By studying these patterns, we can gain a better understanding regarding the underlying coevolutionary dynamics [45] and their effect on shaping specific prey–predator interactions [15]. For example, whether they are idiosyncratic and hard to predict from one ecosystem to another [46]. This is particularly important since the vast majority of phages and microbes are hard to isolate and propagate in the laboratory. Therefore, we studied the structure of the Salmonella phage–bacteria infection network (PBIN) using the available in vitro infection data. A first important observation is that the PBIN is not random, i.e., the pattern of “who-infects-whom” is statistically different from what would be expected if cross-infection occurred by chance. This means that there are factors controlling the outcome of the infection and shape the structure of the network. For example, it is known that cross-infection at the community scale depends on the underlying genetics of host defense and phage counter-defense mechanisms, but also on the ecological context in which evolution acts [10]. Moreover, the PBIN did not have a “one-to-one” organization, suggesting that there is not elevated speciation, such that each phage would only infect one host and each host would only be infected by one phage [10]. If that was the case, in the rearranged Salmonella phage–bacteria infection matrix we would observe a nearly diagonal matrix since phages would infect a unique host or a limited number of closely related host. In contrast, the PBIN was nested when compared to Sim1 and Sim8 null matrices, i.e., the specialist phages infect the most susceptible hosts to infection. As a result of this effect, the PBIN contains interactions that form a hierarchy for both phages and their hosts [10]. Notably, the nested pattern is predicted to arise as a result of gene-for-gene processes [10,47,48], in which acquiring new bacterial mutations confer bacterial resistance to recently evolved phages, while maintaining resistance to past phages. In a similar way, phages acquire mutations expanding their host range without losing their ability to infect ancestral host genotypes [10]. Interestingly, as a result of this process we would expect the co-occurrence of specialist and generalist phage species in the same environmental source rather than the presence of specialist or generalist species alone in specific sites. This was also observed in our study. For example, in the Mtkari river water, we isolated Salmonella phages with a broad range of infectivity spanning from the generalist GEC_vB_B1 (Myoviridae) infecting 111 strains to the GEC_vB_7A (Myoviridae) infecting 52 strains. Or in human isolates we observed bacteria being infected by all 13 phages (T65 S. Enteritidis) but also the hard-to-infect T66 S. Typhimurium (no infections) and T45 S. Typhimurium (infected by three phages). The nested pattern also suggests that the observed host range of the Salmonella phages are subsets of each other and the phage susceptibility of the Salmonella bacteria are also subsets of each other. The gene-for-gene coevolution process also implies that taxonomy is strongly associated with the infection outcome [10,49]. In support of this, Wichels et al. observed a gradient of host ranges from Myoviridae to Siphoviridae to Podoviridae in the North Sea, and that the host ranges of these viruses are nested [50]. This is also what we observed in our study. Myoviridae infected on average much more Salmonella bacteria strains than Siphoviridae, than Podoviridae. Overall, the nested pattern characterizing the PBIN is in line with the weak correlation between the infection profiles of the Salmonella phages and the environmental source or the taxonomic lineage alone, since both generalist and specialist species are present with overlapping target-range. Interestingly, we detected a weak signal of modularity. A modular PBIN contains interactions that occur among distinct groups of phages and hosts and arise as a result of speciation [10]. There are many possible drivers of modularity, including geographic isolation, which can facilitate the divergent coevolution of interacting species [51,52]. In our study we observed that phages GEC_vB_7A and GEC_vB_TR belonged to the same PBIN clique (clique 4). Notably, both viruses were isolated from the same environmental site (Mtkari river), shared similar infection profiles (cluster 2 phages), and presented extremely narrow target range. Therefore, it is possible that these viruses coevolved towards increased speciation. However, in the other PBIN cliques we also observed the co-occurrence of phage species isolated from different environmental sources (e.g., Mtkari and Black Sea). This could happen if different environmental sites share similar microbial isolates but that their geographic separation or different community niches facilitated local coevolution to take place, which enabled divergences in functional interactions [46,53,54]. Overall, nestedness and modularity are not mutually exclusive; there is evidence of trade-offs between the two processes [55,56], and nested patterns could form within modules [10,57,58].

Our results have important implications for Salmonella phage therapy. One of the challenges in phage therapy is the isolation and characterization of phage clones with specific infection profiles. Therefore, understanding the underlying factors that contribute to the phages target preferences is important in creating an arsenal of phages with different target ranges. For example, based on our data, Salmonella phages isolated from different geographical and environmental sources could infect similar bacteria species. Alternatively, phage clones belonging to different taxonomic lineages could infect a variable number of bacteria. Notably, genetically similar phages were also more similar in their infection profiles. All the above can aid in the prediction of the Salmonella phages target-range and help in the design of phage mixtures with different therapeutic capacity. Valuable information can also be obtained from the analysis of the Salmonella PBINs. The observed nested structure suggests the presence of phage clones with a gradient of target-range from generalists to specialists. As we demonstrated, generalists can coexist with specialists in the same environmental site, suggesting that phages with variable infection characteristics can be isolated from the same ecological niche. The nested pattern in combination to the modular can inform strategies to prepare phage cocktails with effectiveness against both a broad range of bacteria species but also towards “hard-to-infect” cases. Finally, in this study we provide an analytical pipeline that can be used in different phage infection datasets, and it can be incorporated and inform in vitro and in vivo experimental processes.

## 5. Conclusions

We characterized the infection profiles and target-range of a set of Salmonella phages against a broad range of Salmonella strains isolated from different environmental sites. These phages may be used for interventions such as therapy, prophylaxis, biocontrol, food, water, and surface decontamination. Within this dataset we identified groups of phages with similar infection profiles and characterized their targets. Genetically similar phages shared similar infection profiles. By studying the structure of the Salmonella phage–bacteria infection network we provide evidence of nonrandom nested organization and weak modularity with important implications for the coevolution mechanisms shaping the ecological interactions between Salmonella phages and their bacterial host.

## Figures and Tables

**Figure 1 viruses-13-01261-f001:**
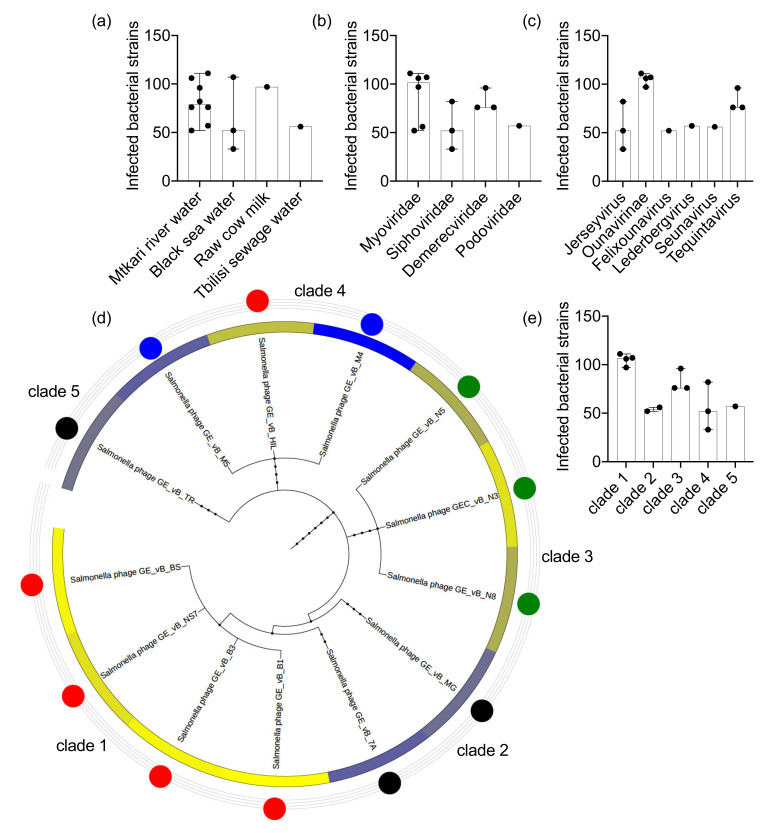
Salmonella phages target-range. The number of phage-infected Salmonella bacteria strains are presented according to the phage (**a**) environmental source, (**b**) taxonomic family, (**c**) taxonomic genus, (**d**) taxonomic lineage, and (**e**) taxonomic clade. The Salmonella phage taxonomic tree in (**e**) includes a circular heatmap which is analogous to the absolute number of infected bacterial strains (log_10_ transformed) and the colored nodes represent the phage infection profile cluster that each phage was assigned to (see Figure 2); red: cluster 1, black: cluster 2, green: cluster 3, and blue: cluster 4.

**Figure 2 viruses-13-01261-f002:**
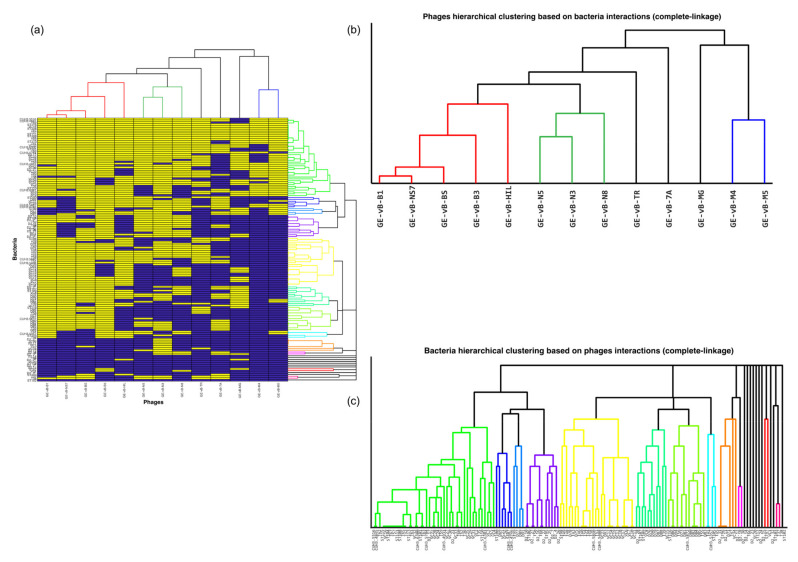
Salmonella phage–bacteria infection profiles and clustering. (**a**) Binary heatmap of the Salmonella phage–bacteria infection matrix representing positive (yellow) and negative (blue) in vitro infections. The matrix is organized according to the unsupervised hierarchical agglomerative clustering of the Jaccard similarity amongst the viral species (horizontal) (**b**) and the bacterial strains (vertical) (**c**). Strains and species with similar infection preferences (profiles) are positioned closer together in the cladograms in a bottom-up organization.

**Figure 3 viruses-13-01261-f003:**
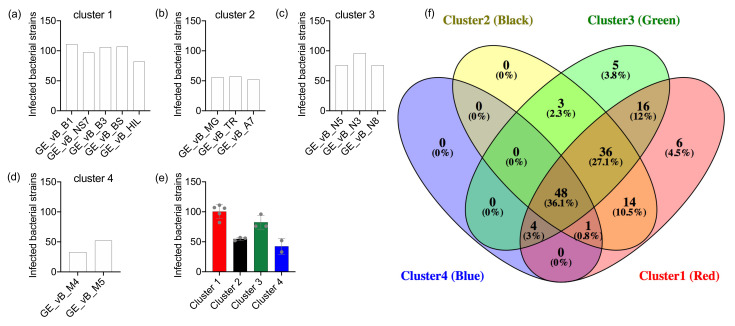
Target-range of Salmonella phages in different phage infection profile clusters. The number of infected Salmonella bacteria strains per each phage are presented for (**a**) cluster 1, (**b**) cluster 2, (**c**) cluster 3, and (**d**) cluster 4. (**e**) The number of infected Salmonella bacteria strains per phage cluster: Median values and 95%CIs are presented. (**f**) Venn diagram of phage-target overlap amongst phage clusters.

**Figure 4 viruses-13-01261-f004:**
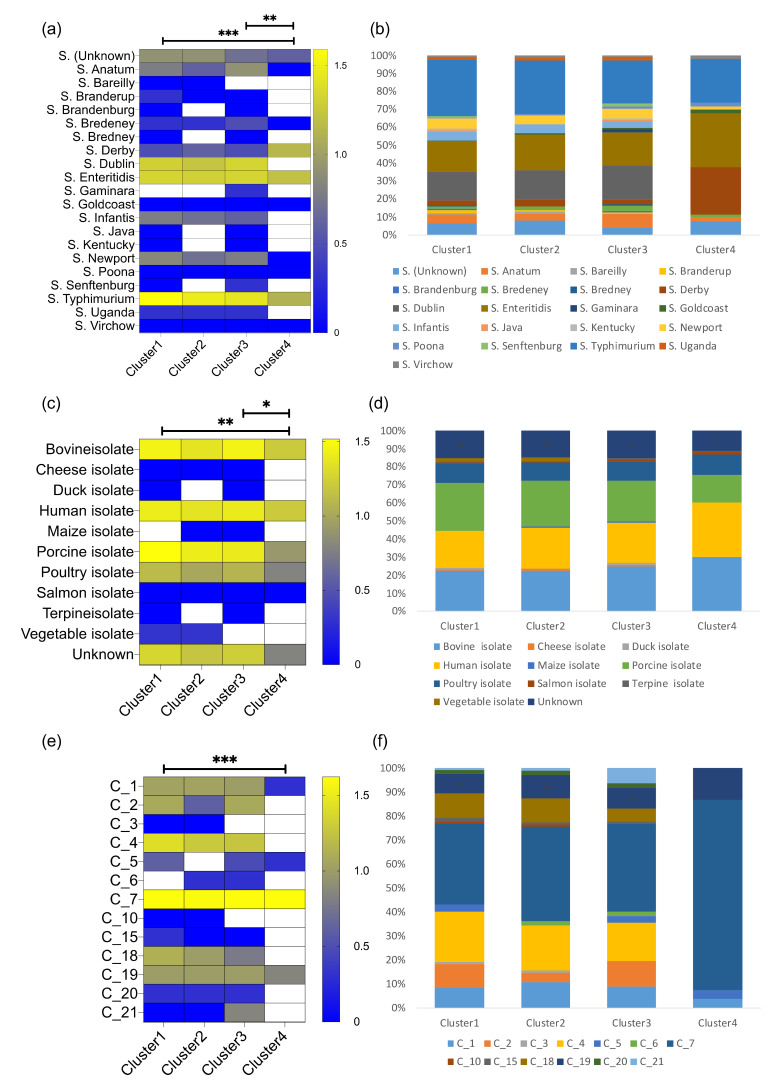
Clusters of Salmonella phage infection profiles. The infection profiles for each Salmonella phage cluster are presented based on three factors: (**a**,**b**) the type of Salmonella bacteria species that were infected, (**c**,**d**) the type of environmental isolates of the bacteria, and (**e**,**f**) the infection cluster that the bacteria belonged to. For each factor the absolute number of infected Salmonella strains is presented in a heatmap format (**a**,**c**,**e**), as well as stacked bar plots representing compositional infection profiles (**b**,**d**,**f**). Heatmap color is analogous to the log_10_ transformed number of absolute counts of successful infections (high—yellow to low—blue). Significant differences were observed amongst clusters 1 and 4, and 3 and 4. Significance tests: * *p* < 0.05, ** *p* < 0.001, *** *p* < 0.0001.

**Figure 5 viruses-13-01261-f005:**
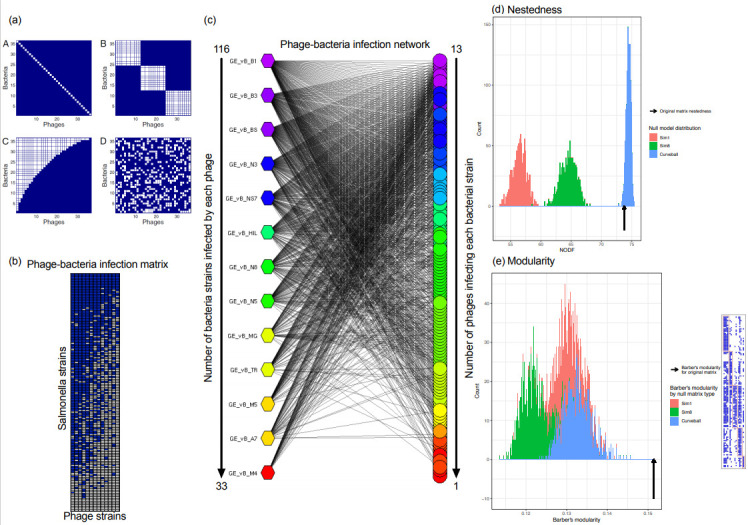
Investigation of the structure of the Salmonella phage–bacteria infection network. (**a**) Expected host–phage interaction matrices (adopted by [8]). Bacteria–phage interactions can be (**a**) A—unique; B—modular; C—nested; D—random. (**b**) The Salmonella phage–bacteria infection matrix (PBIM): Rows—Salmonella bacterial strains arranged with decreasing order based on the number of different phage species that can be infected by. Columns—phage species arranged with decreasing order based on the number of different Salmonella bacterial strains that can infect. Color coding: blue—infection, white—no infection. (**c**) The Salmonella PBIN; the network is arranged in accordance with the data from the PBIM. Circular nodes—Salmonella bacterial strains; polygonal nodes—phage species. Color coding—high (pink) to low (red) number of interactions (infections). Color coding refers to phage and bacterial node degree (number of successful infections), respectively. (**d**) Analysis of nestedness in the Salmonella PBIN. Distributions of nestedness scores (NODF) for the three null models (1200 simulations each) are shown, with the arrow indicating the score for the observed matrix. A NODF score for the observed matrix greater than 95% of the null matrix scores indicates the observed data is significantly nested (*p* < 0.05). Null models: red—Sim1; green—Sim 8; blue—Curveball. (**e**) Analysis of modularity in the Salmonella PBIN. Distributions of modularity scores for the three null models (1200 simulations each) are shown, with the arrow indicating the score for the observed matrix. A modularity score for the observed matrix greater than 95% of the null matrix scores indicates the observed data is significantly modular (*p* < 0.05). Matrix visualization of the module assignment giving the highest modularity score. Red—Sim1; green—Sim 8; blue—Curveball.

**Table 1 viruses-13-01261-t001:** Characteristics of the Salmonella phage species. The table includes metadata on the characteristics of the 13 Salmonella phage species investigated in this study. These include the environmental source, the taxonomic clustering at the family and genus level and the phylogenetic clade, the infection profile cluster, and the target range.

Source	Name	Study id	Tax id	Family	Genus	Taxonomic Clade	Infection Profile	Infected Hosts
Mtkari river water	Salmonella phage GEC_vB_B1	GE_vB_B1	2108164	Myoviridae	Unclassified Ounavirinae	Clade 1	Cluster 1	111
	Salmonella phage GEC_vB_B3	GE_vB_B3	2108165	Myoviridae	Unclassified Ounavirinae	Clade 1	Cluster 1	106
	Salmonella phage GEC_vB_HIL	GE_vB_HIL	2108167	Siphoviridae	Unclassified Jerseyvirus	Clade 4	Cluster 1	82
	Salmonella phage GEC_vB_7A	GE_vB_7A	2108163	Myoviridae	Unclassified Felixounavirus	Clade 2	Cluster 2	52
	Salmonella phage GEC_vB_TR	GE_vB_TR	2108174	Podoviridae	Unclassified Lederbergvirus	Clade 5	Cluster 2	57
	Salmonella phage GEC_vB_N5	GE_vB_N5	2108171	Demerecviridae	Unclassified Tequintavirus	Clade 3	Cluster 3	76
	Salmonella phage GEC_vB_N8	GE_vB_N8	2108172	Demerecviridae	Unclassified Tequintavirus	Clade 3	Cluster 3	76
	Salmonella phage GEC_vB_N3	GEC_vB_N3	2777377	Demerecviridae	Unclassified Tequintavirus	Clade 3	Cluster 3	96
Black sea water	Salmonella phage GEC_vB_BS	GE_vB_BS	2108166	Myoviridae	Unclassified Ounavirinae	Clade 1	Cluster 1	107
	Salmonella phage GEC_vB_M4	GE_vB_M4	2108168	Siphoviridae	Unclassified Jerseyvirus	Clade 4	Cluster 4	33
	Salmonella phage GEC_vB_M5	GE_vB_M5	2108169	Siphoviridae	Unclassified Jerseyvirus	Clade 4	Cluster 4	52
Raw cow milk	Salmonella phage GEC_vB_NS7	GE_vB_NS7	2108173	Myoviridae	Unclassified Ounavirinae	Clade 1	Cluster 1	97
Tbilisi sewage water	Salmonella phage GCE_vB_MG	GE_vB_MG	2108170	Myoviridae	Unclassified Seunavirus	Clade 2	Cluster 2	56

## Data Availability

The data and the code used in this study are available in the Appendix A.

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
