# Peer review of "Investigation of Salmonella Phage–Bacteria Infection Profiles: Network Structure Reveals a Gradient of Target-Range from Generalist to Specialist Phage Clones in Nested Subsets"

_viruses, 2021, doi:10.3390/v13071261_

Round 1

Reviewer 1 Report

This a solid paper, describing the bacteria-phage networks of Salmonella and salmonella phage.  This is key information for Phage therapy.

  1. The paper is well written, although it could be substantially shortened in places. Summarising the results in particular and using sub-headings would make it much easier to follow.  
  2. That said, the work could do with more discussion about implications for phage therapy. For example, modularity highlights value of carefully selecting components of phage cocktails.
  3. It would be nice to see some more formal analyses of one of the key findings: phage infection profile is not strongly predicted by phage taxonomy. This could be done very simply using Mantel tests, by determining how pairwise phage genetic distance predicts pairwise Jaccard distances.
  4. Phage are fairly genetically distinct so the interaction networks are may have little to do with coevolution: some bacteria may simply lack specific target receptors. The cited articles (57,58) are about very rapid coevolution (over days) shaping the interaction network

Author Response

This a solid paper, describing the bacteria-phage networks of Salmonella and salmonella phage.  This is key information for Phage therapy.

Response: We would like to thank Reviewer 1 for the constructive comments. Especially, for the suggestion to perform the Mantel tests. We have addressed all comments.

Point 1: The paper is well written, although it could be substantially shortened in places. Summarising the results in particular and using sub-headings would make it much easier to follow.  

Response 1: We have shortened the text, where possible, and we have summarised the results in the following subheadings.

“The Salmonella phage target-range varies between clones of different taxonomic lineages and isolated from different environmental sites.” (line 229)

“The Salmonella phages are grouped into clusters with different infection profiles”. (line 257)

“The genetic distances of the Salmonella phages are positively associated to their infection profiles”. (line 331)

“The Salmonella phage-bacteria infection network is nested”. (line 346)

Point 2: That said, the work could do with more discussion about implications for phage therapy. For example, modularity highlights value of carefully selecting components of phage cocktails.

Response 2: We have added a paragraph in discussion discussing possible implication for phage therapy; lines 520-538

Point 3: It would be nice to see some more formal analyses of one of the key findings: phage infection profile is not strongly predicted by phage taxonomy. This could be done very simply using Mantel tests, by determining how pairwise phage genetic distance predicts pairwise Jaccard distances.

Response 3: We have performed the Mantel test with 10,000 permutations. We observed a possible association between the genetic distances and the Jaccard distances. The method is described in the methodology section (lines 213-215). The results are discussed in the results section (lines 331-343), and presented in Supplementary Figure 3.

Point 4: Phage are fairly genetically distinct so the interaction networks are may have little to do with coevolution: some bacteria may simply lack specific target receptors. The cited articles (57,58) are about very rapid coevolution (over days) shaping the interaction network

Response 4: We have rephrased (line 518)

Reviewer 2 Report

The manuscript by Makalatia et al characterize 13 phage clones against 141 Salmonella strains.  From these data, they apply a Salmonella phage-bacterial infection network from which they derive clusters to examine the relationship/ associations between the various phages.   

MINOR EDITORIAL:

  1. Line 34: Better word choice may be “people becoming ill”.
  2. Line 54: Should be “prey” not “pray”.
  3. Line 122: What is the media for growing o/n bacterial cultures, LB broth?
  4. Line 126: Should the titer be 10^8 CFU/ ml no 108 CFU/ml???

MAJOR:

Unfortunately, the authors do not have an actual “take away” message from this manuscript, which is a major weakness.  The authors state there is no clear association between any of the characteristics studied.  The methods appear to be done correctly, there is just no correlation with the samples tested.  Perhaps this manuscript would read better if it was provided as a methods paper?

Author Response

The manuscript by Makalatia et al characterize 13 phage clones against 141 Salmonella strains.  From these data, they apply a Salmonella phage-bacterial infection network from which they derive clusters to examine the relationship/ associations between the various phages.   

Response: We would like to thank Reviewer 2 for the constructive comments. We have addressed all of them.

MINOR EDITORIAL:

  1. Line 34: Better word choice may be “people becoming ill”.
  2. Line 54: Should be “prey” not “pray”.
  3. Line 122: What is the media for growing o/n bacterial cultures, LB broth?
  4. Line 126: Should the titer be 10^8 CFU/ ml no 108 CFU/ml???

 Responses:

  1. Line 39: “infected”
  2. Line 67: “prey”
  3. Line 137: “LB broth”
  4. Line 141: “108”

MAJOR:

Unfortunately, the authors do not have an actual “take away” message from this manuscript, which is a major weakness.  The authors state there is no clear association between any of the characteristics studied.  The methods appear to be done correctly, there is just no correlation with the samples tested.  Perhaps this manuscript would read better if it was provided as a methods paper?

Response: Our study has four take away messages which have been emphasised throughout the revised manuscript and are summarised as subheading in the results sections:

 “The Salmonella phage target-range varies between clones of different taxonomic lineages and isolated from different environmental sites.” (line 229)

“The Salmonella phages are grouped into clusters with different infection profiles”. (line 257)

“The genetic distances of the Salmonella phages are positively associated to their infection profiles”. (line 331)

“The Salmonella phage-bacteria infection network is nested”. (line 346)

We have also included a paragraph in the discussion section to relate our findings with phage therapy (lines 520-538)

Round 2

Reviewer 2 Report

My comments/ concerns have been addressed.